TOPICAL REVIEW

# Enviromimetics: From exercise mimetics to cognitomimetics in the quest for enhanced brain health and cognition

Anthony J. Hannan[1,2,3] 

[1] *Florey Institute of Neuroscience and Mental Health, Parkville, Australia*
[2] *Florey Department of Neuroscience and Mental Health, University of Melbourne, Parkville, Australia*
[3] *Department of Anatomy and Physiology, University of Melbourne, Parkville, Australia*

Handling Editors: Laura Bennet & Rebecca MacPherson

The peer review history is available in the Supporting Information section of this article (https://doi.org/10.1113/JP287484#support-information-section).

**Abstract figure legend** A schematic diagram outlining the concept of enviromimetics, and the subclasses of exercise mimetics, epimimetics and cognitomimetics. The rationale is that environmental stimulation and lifestyle factors, including physical activity and cognitive stimulation, have shown beneficial effects across a range of human conditions, including a wide variety of neurological and psychiatric disorders. Enviromimetics are novel therapeutics which mimic or enhance the beneficial effects of environmental stimulation. Exercise mimetics are a subclass of enviromimetics which mimic or enhance the beneficial effects of physical activity. Cognitomimetics are a new subclass of enviromimetics, proposed for the first time in this article, that mimic or enhance the beneficial effects of cognitive stimulation. Epimimetics are a subclass of enviromimetics which specifically target epigenetic modifications resulting from therapeutic environmental stimulation. These novel therapeutics will target molecular mediators of these forms of experience-dependent plasticity. Each type of enviromimetic will require systematic testing and optimisation in valid preclinical models, prior to translation.

**Abstract** Enviromimetics were first proposed over two decades ago, as novel therapeutics to mimic or enhance the beneficial effects of environmental stimulation. In the intervening period, subclasses of enviromimetics have been proposed, most notably exercise mimetics. Epimimetics constitute an additional subclass of enviromimetics, which act via epigenetic mechanisms. In this article, the concept of enviromimetics is updated, including its subclasses, and explored in the context of the development of novel therapeutic approaches to a wide range of human disorders, with a specific focus on neurological diseases and psychiatric disorders. Furthermore, a new concept is introduced, that of 'cognitomimetics', which specifically mimic or enhance the therapeutic effects of cognitive stimulation. One focus of discussion is the beneficial molecular and cellular mechanisms induced by environmental exposures and lifestyle factors, including increased physical activity and cognitive stimulation. Exercise mimetics represent the largest, and most experimentally tractable, subclass of enviromimetics, due to the biologically pervasive and readily quantifiable therapeutic impacts of physical activity, both within the nervous system, and throughout the body. These mechanisms provide new insights into molecular targets for these novel therapeutic approaches. It is hoped that this will lead to new ways to prevent, ameliorate and eventually cure a wide range of human illnesses, particularly brain disorders, which collectively constitute the largest, and most rapidly growing, global burden of disease.

(Received 2 March 2025; accepted after revision 3 September 2025; first published online 25 September 2025)

**Corresponding author** A.J. Hannan: Florey Institute of Neuroscience and Mental Health, University of Melbourne, Parkville VIC 3010, Australia. Email: anthony.hannan@florey.edu.au

## Introduction

Neurological and psychiatric disorders collectively represent the largest, and most rapidly growing, burden of disease. However, most individuals suffering from the majority of these major disorders remain untreated, or inadequately treated. There is therefore an enormous unmet clinical need in the fields of neurology and psychiatry, as well as other areas of medicine where neurological and psychiatric comorbidities are prevalent. Addressing these challenges requires new approaches, based on mechanistic insights across neuroscience and related fields.

One approach is to develop new therapeutics based on non-drug interventions that have been found to have preventative and/or therapeutic effects. The concept of enviromimetics was first proposed over two decades ago (Hannan, 2004) and has been elaborated upon in subsequent publications (Hannan, 2014; McOmish & Hannan, 2007). Enviromimetics are novel therapeutics which mimic or enhance the beneficial effects of environmental stimulation, including physical activity and cognitive stimulation. Exercise mimetics are a subclass of enviromimetics (Gubert & Hannan, 2021; Narkar et al., 2008). Exercise mimetics mimic or enhance the therapeutic effects of physical activity (Gubert & Hannan, 2021).

In this article, I propose an additional subclass of enviromimetics: 'cognitomimetics'. Cognitomimetics are therapeutics which mimic or enhance the beneficial effects of cognitive stimulation. Cognitive stimulation is described in the literature in various contexts including 'mental stimulation', 'mental activity' and 'cognitive activity'. There is accumulating evidence that cognitive stimulation can enhance brain health and slow brain ageing and neurodegeneration (Grande et al., 2020). It is proposed that the development of cognitomimetics would complement that of exercise mimetics, and both would be therapeutic subclasses of enviromimetics. Before proceeding to discuss cognitomimetics, and other subclasses of enviromimetics, in more detail, I will first

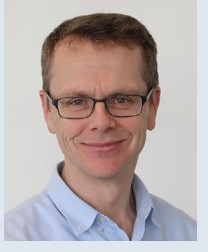

**Anthony J. Hannan** is a professor at the Florey Institute of Neuroscience and Mental Health, University of Melbourne. Hannan and colleagues provided the first demonstration in any genetic animal model that environmental stimulation can be therapeutic. This has led to new insights into gene–environment interactions in various brain disorders, including Huntington's disease, dementia, autism, schizophrenia, depression and anxiety disorders. Recent discoveries extend to epigenetic inheritance and the microbiota–gut–brain axis as modulators of cognitive and affective function and dysfunction.

address the general concept and potential utility of enviromimetics.

## Enviromimetics

Enviromimetics are a broad class of therapeutics which were initially inspired by the literature on the beneficial effects of environmental enrichment (Hannan, 2004; McOmish & Hannan, 2007). Environmental enrichment, as it applies to laboratory animals (including preclinical models) involves increasing novelty and complexity in home environments so as to enhance sensory stimulation, cognitive exercise and opportunities for physical activity (Gubert & Hannan, 2019; Mo et al., 2016).

Environmental enrichment has shown therapeutic effects in delaying onset and/or slowing progression in preclinical models of a wide range of neurological and psychiatric disorders (Nithianantharajah & Hannan, 2006; Renoir et al., 2013; Rogers, Renoir et al., 2019). The logic behind the concept of enviromimetics is that harnessing the beneficial effects of environmental enrichment and related forms of environmental stimulation can identify molecular targets for this novel class of therapeutics (Hannan, 2004).

In the past two decades, the greatest progress has been made with a specific subclass of enviromimetics, exercise mimetics (Gubert & Hannan, 2021). This is partly because physical exercise, including stimulation of muscles and the cascading effects across most cells and tissues in the body, is highly quantifiable and mechanistically tractable. The other reason is that physical activity has been found to have beneficial effects on more human disorders than almost any other lifestyle factor (with the possible exception of diet and nutrition). Therefore, exercise mimetics are at the core of this article, and will be discussed in detail below.

## Exercise mimetics

Physical exercise is a lifestyle factor with almost unparalleled beneficial impacts on a range of human disorders, including neurological and psychiatric disorders (Fuza-Luces et al., 2018; Febbraio, 2017; Gleeson et al., 2011; Hillman et al., 2008; McTiernan, 2008). Physical activity is intrinsic to human behaviour and our species has evolved to be highly active, whether hunting, gathering or engaging in the multitude of activities that maintain and optimise life. Conversely, this means that modern lifestyles associated with sedentary behaviours (and poor-quality nutrition) can make individuals prone to a wide range of disorders of body and brain.

As mentioned above, the core concept of exercise mimetics is that we can harness the beneficial effects of physical activity in order to develop novel therapeutics. In other words, exercise mimetics mimic or enhance the beneficial effects of physical activity (Carey and Kingwell, 2009; Fan et al., 2013; Fan and Evans, 2017; Gubert & Hannan, 2021; Narkar et al., 2008; Wall et al., 2016). It is proposed that mimicking or enhancing the therapeutic impacts of exercise will constitute new approaches to delay or ameliorate a range of human disorders. Exercise mimetics have been reviewed previously (e.g. Gubert & Hannan, 2021; Hawley et al., 2021). Therefore, readers are directed to these reviews for extensive discussion of this class of therapeutics. In the present article, the general concept of exercise mimetics, and progress in recent years, will be discussed.

The potential of exercise mimetics for the prevention and treatment of a variety of disorders is increasingly recognised, and brain disorders constitute a major target. This includes a range of neurological disorders (McDonnell et al., 2011), most notably Alzheimer's disease (Cammisuli et al., 2018; Farina et al., 2014; Strohle et al., 2015), other dementias (Li et al., 2019), Parkinson's disease (Cruickshank et al., 2015; Shu et al., 2014; Uhrbrand et al., 2015), amyotrophic lateral sclerosis (Meng et al., 2020), multiple sclerosis and stroke (Hung et al., 2021; Luo et al., 2020; Vanderbeken and Kerckhofs, 2017). The therapeutic potential of physical activity extends to effects on cognitive function during maturation (Alvarez-Bueno et al., 2017) and ageing (Barha et al., 2017; Calverley et al., 2020; Northey et al., 2018). Furthermore, major psychiatric disorders, such as depression (Dauwan et al., 2016) and schizophrenia (Dauwan et al., 2016; Firth et al., 2015, 2016), may be responsive to exercise mimetics.

There are various demonstrations of how exercise might increase the health of specific organs, and the brain in particular. For example, transfer of circulating blood factors from aged mice exposed to an exercise intervention, *versus* a non-exercise control, demonstrated transfer of the beneficial effects of physical exercise on cognition, as well as a cellular correlate of specific cognitive (learning and memory) processes, adult hippocampal neurogenesis (Horowitz et al., 2020). In this key study, the circulating blood factors from exercised aged mice were administered to sedentary aged mice, demonstrating enhancement of specific aspects of brain health. These investigators implicated glycosylphosphatidylinositol (GPI)-specific phospholipase D1 (Gpld1), a GPI-degrading enzyme derived from liver, in these beneficial effects, and thus the liver–brain axis (Horowitz et al., 2020). Below, I will also outline a range of molecular targets for exercise mimetics, as well as potential mechanisms that may inform therapeutic optimisation.

**Myokines as molecular targets of exercise mimetics.** One major group of candidate molecular targets for

exercise mimetics are myokines. Myokines are released from muscles and can signal to other cells and organs throughout the body, including the brain. The exact signals associated with induction of myokine release following exercise are not well understood, but presumably start with the physiological and metabolic processes associated with increased muscle usage (Bigliassi et al., 2026; Murphy et al., 2020; Pedersen & Febbraio, 2012; Zare et al., 2025).

Various myokines have been identified, including irisin, interleukin-6 (IL-6), lactate, insulin-like growth factor 1 (IGF-1) and cathepsin B. Following their release from muscles, myokines may modulate the function of various cells, organs and biological systems throughout the body. This includes impacts on the brain (via the blood–brain barrier), such as modulation of synaptic plasticity and adult neurogenesis, and consequent induction of cognitive enhancement and antidepressant-like effects (Kim et al., 2019).

Myokines appear to be one of the earliest, and most important, mediators of signalling from muscles to other organs (including the brain) following exercise (Pedersen, 2011; Whitham and Febbraio, 2016). The kinds of myokines that are released from muscles as signalling molecules are thought to include various cytokines (e.g. IL-6), brain-derived neurotrophic factor (BDNF), irisin, meteorin-like protein (Metrn1), IGF-1, $\beta$-aminoisobutyric acid (BAIBA), lactate and cathepsin B (Moon et al., 2016). With respect to cognitive enhancement following myokine secretion, key molecular mediators implicated include IL-6, BDNF, irisin, cathepsin B, lactate and kynurenine acid (Kim et al., 2019; Yu et al., 2025).

The first myokine to be described and extensively studied was IL-6 (Fiuza-Luces et al., 2018; Pedersen & Febbraio, 2005; Pedersen et al., 2004). The IL-6 cytokine was first studied as an inflammatory mediator. The release of IL-6 from skeletal muscle following physical exercise raised possibilities of a broader function as a myokine. Production of IL-6 in myocytes has been found to require the $Ca^{2+}$–NFAT (nuclear factor of activated T cells) and glycogen–p38 MAPK (mitogen-activated protein kinase) signalling pathways. The evidence that IL-6 release is not dependent on prior TNF-alpha release has suggested a potential metabolic role for IL-6. Furthermore, IL-6 has been implicated in adult neurogenesis (Bowen et al., 2011; Vallieres et al., 2002). Finally, IL-6 was functionally linked to appetite regulation via neuropeptides and hypothalamically mediated energy homeostasis (Pedersen, 2019; Solmi et al., 2020).

Another myokine induced by exercise, and thus of relevance to exercise mimetics, is IGF-1 (Carro et al., 2000; Nakajima et al., 2010). Exercise-induced IGF-1 has been found to upregulate Akt-CREB-mediated hippocampal BDNF levels, which enhances forms of cellular plasticity including adult neurogenesis and synaptic plasticity (Ding et al., 2006). Blocking IGF-1 can abrogate this response.

A further exercise-induced myokine of interest is cathepsin B which is also associated with upregulated hippocampal BDNF expression and adult neurogenesis. These molecular and cellular impacts of cathepsin B, following its release from muscles as a myokine, may in turn enhance cognitive performance (Moon et al., 2016).

Lactate, a metabolite produced when muscles contract, has also involved been implicated as a myokine and BDNF upregulator (Sleiman et al., 2016). Increased lactate levels correlated with upregulated BDNF in the periphery (Schiffer et al., 2011). Transport of lactate across the blood–brain barrier involves monocarboxylate transporters. The subsequent upregulation of BDNF expression is thought to require silent information regulator 1 (SIRT1) and the PGC1$\alpha$/FNDC5/BDNF pathway. The molecular effects of lactate upregulation may involve selective transcription factors regulating expression of genes involved in synaptic plasticity and other forms of neural plasticity, as well as cognitive performance (El Hayek et al., 2019).

Irisin (also known as FNDC5) is another key myokine induced by exercise and downstream of the transcriptional coactivator PGC1$\alpha$. Irisin is another myokine that may act via upregulation of brain BDNF expression (Wrann et al., 2013). The beneficial effects of irisin may involve forms of cellular plasticity, including synaptic plasticity (Kim et al., 2019). One key impact of irisin may be cognitive enhancement, with relevance for Alzheimer's disease, other forms of dementia, and other cognitive disorders (Lourenco et al., 2019).

Overall, there appear to be many effects of myokines in brain function, and dysfunction (Chow et al., 2022; de Freitas et al., 2020; Isaac et al., 2021; Kim et al., 2022; Marcourt et al., 2025; Matthews et al., 2023; Voss et al., 2019; Yu et al., 2025). Each of these myokines has different molecular characteristics. Therefore, targeting individual myokines for therapeutic applications requires an understanding of the relevant molecular biology and pharmacology. This may include pharmacological targeting with small-molecular drugs, as well as biologicals and gene-therapy approaches (Gubert & Hannan, 2021).

**Beyond myokines: exerkines as therapeutic targets for exercise mimetics.** Exerkines have been defined as 'signalling moieties released in response to acute and/or chronic exercise, which exert their effects through endocrine, paracrine and/or autocrine pathways' (Gerszten et al., 2022). Therefore, whilst myokines (which are released from skeletal muscles; Pedersen et al., 2004) are the largest and most well studied subclass of exerkines (as discussed above), other exerkines include cardiokines

(released by the heart; Doroudgar and Glembotski, 2011), adipokines (released by white adipose tissue; Pedersen & Febbraio, 2005), batokines (released by brown adipose tissue) and hepatokines (release by the liver; Hansen et al., 2011).

The definition of these multiple classes of exerkines draws attention to the complexity of exercise biology. Whilst physical exercise starts with increased activity of skeletal and cardiac muscles, it rapidly encapsulates most (if not all) major organs in the body, and signalling between many of these organs is bidirectional, and highly spatiotemporally regulated. We are only beginning to understand how these different organs and systems signal to each other during, and after, exercise. More detailed exploration of molecular and cellular mechanisms will help to identify the subset of exerkines which may provide the most promising candidates as molecular targets of exercise mimetics. Nevertheless, exerkines (including many myokines and the other subclasses discussed above) are only one class of molecules mediating the effects of physical exercise. Below, other candidate molecular targets for exercise mimetics will be discussed.

**Other molecules that are candidate targets for exercise mimetics.** Metabolic regulating pathways that are proximal to the direct effects of exercise on muscles include the AMPK-Sirtuin 1 (SIRT1)-peroxisome proliferator-activated receptor-$\gamma$ coactivator (PGC-1-PPAR$\delta$) pathway (Hoffman, 2017; Blazev et al., 2022). This has been actively explored in the context of exercise mimetics (Fan and Evans, 2017). Therapeutics targeting components of this pathway may include 5-aminoimidazole-4-carboxamide ribonucleotide (AICAR), metformin (with AMPK as a target) and GW501516 (with PPAR$\delta$ as a target) (Kobilo et al., 2011, 2014; Fan and Evans, 2017; Hervas et al., 2017; Lauritzen et al., 2013; Moon et al., 2016). The beneficial effects of AICAR were most notable, and included enhanced hippocampal neurogenesis and improved cognitive (spatial learning and memory) and motor performance in mice (Kobilo et al., 2011, 2014). It is possible that the observed increase in hippocampus neurogenesis was directly responsible for the enhanced spatial learning and memory; however, a causal link was not demonstrated with respect to the mechanisms of AICAR action (Kobilo et al., 2011).

However, whilst targeting the AMPK-SIRT1-PGC1-PPAR pathway remains a promising strategy for a variety of brain disorders, the broad function of the pathway also suggests that interventions (including those repurposed from other indications) may have significant side-effect profiles, and require pharmacological optimisation (Birajdar et al., 2023; Rakshe et al., 2024). Nevertheless, knowledge of these pathways activated by exercise can provide insight into potential targets for exercise mimetics, which may include small molecules, large molecules (e.g. biologics) and gene-therapy tools.

Molecules that are more distal to the direct effects of exercise on muscle, but may nevertheless mediate the effects of exercise on brain function and cognition, are thought to include the neurotrophin BDNF and its main receptor TrkB (Ieraci et al., 2016; Neeper et al., 1996; Pang et al., 2006; Park et al., 2019; Seo et al., 2019; Vaynman et al., 2004; Venezia et al., 2016). As discussed above, BDNF has been extensively studied with respect to the experience-dependent neural effects of cognitive stimulation, physical activity, and their combination (environmental enrichment). Whilst one might suggest that some of this has been due to 'looking under the lamp-post' (known molecules are most likely to be investigated in follow-up studies), the evidence supporting causative roles of BDNF signalling (as discussed elsewhere in this review) is substantial (Kim et al., 2024; Mohandas et al., 2025). In addition to evidence from preclinical models, there is independent evidence supporting a role for BDNF genetics in human brain development and experience-dependent plasticity (Mohandas et al., 2025; Notaras & van den Buuse, 2020; Zarza-Rebollo et al., 2024).

Furthermore, a range of glutamatergic, serotonergic, dopaminergic and adrenergic signalling molecules, and their associated receptor-mediated signalling pathways, have been implicated in exercise-induced enhancement of brain function (Anacker and Hen, 2017; Garcia et al., 2017; Graff et al., 2018; Kim et al., 2018; Klempin et al., 2013; Parrini et al., 2017; Reichmann et al., 2016; Rogers, Renoir et al., 2019; Shi et al., 2019). Other molecular mediators of exercise effects on the brain may include specific neuropeptides such as neuropeptide Y (NPY) and their receptors and associated signalling cascades (Chen et al., 2007; Jensen et al., 1994; Joksimovic et al., 2017; Ramson et al., 2012; Reichmann et al., 2016).

Another key class of molecular mediators of the beneficial effects of exercise include small non-coding RNAs (sncRNAs), which provide a variety of therapeutic targets. The subclasses of sncRNAs that have been best studied are microRNAs. A variety of sncRNAs, including microRNAs, are modulated by exercise (e.g. miR-29a-3p, miR-200a-3p and miR-204; Kuznetsova et al., 2024; Lee et al., 2025; Pinto-Hernandez et al., 2025; Shima et al., 2025). The other major class of non-coding RNAs are long non-coding RNAs (lncRNAs). Various lncRNAs have also been found to be regulated by exercise (e.g. CYTOR and Tug1; Bonilauri and Dallagiovanna, 2020; Han et al., 2025; Trewin et al., 2022; Wohlwend et al., 2021) and thus may also be targets for exercise mimetics.

The fact that each microRNA can bind the 3'UTR of multiple mRNA targets allows them to be mapped

onto specific regulatory networks and associated with functional outcomes such as cognitive enhancement (Fernandes et al., 2018). Specific microRNAs have been implicated in the exercise-induced upregulation of adult neurogenesis and associated changes in neural function (Kuznetsova et al., 2024; Pons-Espinal et al., 2019). Thus, the modulation of one or more microRNAs by a physical exercise intervention (and associated stimulatory interventions such as environmental enrichment) can potentially have substantial and multigenic molecular effects (Kuznetsova et al., 2020).

The microRNA-mediated effects of exercise were also seen in a preclinical model of traumatic brain injury (Bao et al., 2014). Further evidence linked these changes to exercise-induced cognitive enhancement, thus identifying an additional candidate target for exercise mimetics (Hu et al., 2015). Furthermore, the impacts of exercise on microRNA-mediated regulatory networks may also occur at the level of miRNA processing, including the Dicer and Exportin molecules (Garner et al., 2020).

Beyond non-coding RNAs, there are various other epigenetic mediators implicated in the therapeutic effects of exercise (Fernandes et al., 2017). These effects include the epigenetic regulation of BDNF transcription via molecular mediators such as histone H3 acetylation levels (Ieraci et al., 2015). A potential role of histone deacetylation (HDACs) has been implicated in research involving the use of HDAC inhibitors (sodium butyrate and valproate) (Ieraci et al., 2015). Epigenetic modulation of *BDNF* gene may be a key component of the long-term benefits of exercise on the brain (Chen et al., 2007; Chen et al., 2009; Kiuchi et al., 2012; Müller et al., 2020). Other epigenetic mediators of the therapeutic effects of exercise may include H4 acetylation and DNMT3b (Cechinel et al., 2016).

One further molecular mediator of exercise-induced effects on the brain, including the enhancement of adult hippocampal neurogenesis, is VEGF (Fabel et al., 2003). These neural effects VEGF may also be relevant to depression effects (Kiuchi et al., 2012). Further evidence has been reported implicating VEGF signalling and related pathways, particular in the context of the beneficial effects of exercise in stroke models (Gao et al., 2014; Pang et al., 2017).

**Cellular mediators of the therapeutic effects of exercise.** Cellular mediators of the proposed beneficial effects of exercise mimetics on the brain may include experience-dependent forms of cellular plasticity, in particular synaptic plasticity and adult neurogenesis (Mo et al., 2015; Nithianantharajah & Hannan, 2006). These types of neural plasticity have been associated with cellular mechanisms mediating specific aspects learning

and memory, as well as other experience-dependent cognitive processes, and hence exercise mimetics may have therapeutic potential as cognitive enhancers (Alkadhi, 2018; Guerrieri et al., 2017).

The existence of adult neurogenesis in various regions of the healthy adult human brain has at times been controversial, but there is nevertheless substantial evidence that it occurs in the dentate gyrus of the hippocampus and the subventricular zone (a source of adult-born neurons in the olfactory bulb; Rendeiro and Rhodes, 2018). The role of adult hippocampal neurogenesis is thought to include pattern separation associated with learning and memory, which may impact on other cognitive and affective processes. The adult-born cells in the subventricular zone translocate along the rostral migratory stream and then mature into neurons of the olfactory bulb, with associated olfactory functions. Specific molecular mediators of adult neurogenesis have also been implicated in exercise-mediated upregulation of neurogenesis (Kozareva et al., 2018).

Cellular processes implicated in these exercise effects, beyond adult neurogenesis, include synaptogenesis, gliogenesis and angiogenesis. Exercise has been found to enhance cellular plasticity, particularly with respect to the structure and function of the hippocampus (Codd et al., 2020; Islam et al., 2020; Nauer et al., 2020).

One other biological system that may mediate therapeutic effects of exercise is the microbiota–gut–brain axis (Cryan et al., 2019; Gubert et al., 2020). Exercise has been shown to enhance gut health, the diversity of the gut microbial community, release of beneficial short-chain fatty acids (particularly butyrate), and the abundance of specific commensal microbes (mainly bacteria) that are considered to be healthy (Allen, Mailing, Niemiro et al., 2018; Allen, Mailing, Cohrs et al., 2018; Campbell et al., 2016; Dalton et al., 2019; Gubert et al., 2020; Mitchell et al., 2018; Monda et al., 2017; Scheiman et al., 2019; Yu et al., 2024). The beneficial effects of exercise on gut microbiota are also thought to be particularly relevant to neurodegenerative diseases and cognitive enhancement (Abraham et al., 2019; Gubert et al., 2020; Kang et al., 2014).

Further cellular mediators of the beneficial effects of exercise, may include extracellular vesicles (EVs) and their molecular cargo. Exercise can greatly increase the numbers of EVs in circulation, allowing exercise to positively influence a wide range of cells, tissues and organs, including the brain (Fruhbeis et al., 2015; Safdar et al., 2016; Whitham et al., 2018). These EV-mediated therapeutic effects of exercise may also be relevant to stroke, Alzheimer's disease and other forms of dementia, based on preclinical findings (e.g. Fuller et al., 2025; Wang et al., 2020).

## Cognitomimetics

Thus far, enviromimetics and their subclass exercise mimetics have been discussed in detail. I now propose the development of another subclass of enviromimetics, to be called 'cognitomimetics'. Cognitomimetics would be novel therapeutics which mimic or enhance the beneficial effects of cognitive stimulation. The fact that cognitive stimulation acts specifically on neural circuits within the brain implies that cognitomimetics would have brain-specific effects, which might reduce their side effects. One of the reasons for this could be that therapeutic targets for cognitomimics are more likely to have expression patterns that are restricted to the nervous system. However, this speculation is predominantly based on the knowledge that many brain-expressed genes are spatially constrained in their expression patterns. In attempting to develop cognitomimetics, each therapeutic target will need to be systematically assessed and further investigated, based on its own merits. Furthermore, the fact that physical exercise has very different molecular and cellular effects to cognitive stimulation, means that exercise mimetics and cognitomimetics are predicted to have distinct molecular targets.

It has been demonstrated, in various preclinical and clinical studies, that cognitive stimulation (and related concepts of 'cognitive training', 'cognitive activity/exercise' and 'mental activity/exercise') can have beneficial effects on a range of brain disorders (Chartier et al., 2024; Rogers, Renoir et al., 2019). However, identifying candidate molecular targets for cognitomimetics is likely to prove more challenging than exercise mimetics. This is because physical exercise is easily quantified and studied, both preclinically and clinically. However, cognitive stimulation is more challenging to investigate, as it involves direct comparison with appropriate control conditions ('no cognitive stimulation') and the neural circuit's activity during different types of cognitive stimulation is complex and heterogeneous.

One further possibility is that the therapeutic effects of exercise mimetics and cognitomimetics could be combined, and might even be synergistic rather than simply additive. The logic behind this hypothesis is that interventions involving environmental enrichment have generally had greater impact than separate exercise or cognitive-stimulation interventions, although they have rarely been systematically compared within the same studies (Liew et al., 2022; Nithianantharajah & Hannan, 2006; Novati et al., 2022; Vaquero-Rodriguez et al., 2023). Regardless, the development of cognitomimetics as a new class of enviromimetics would allow such hypotheses to be tested, and such novel therapeutic approaches to be optimised.

One challenge in this field is definition of 'cognitive stimulation' in humans, and how it might best be modelled in experimental animal species, such as laboratory mice and rats. In humans, cognitive stimulation may involve learning new skills (mental and physical), engaging in cognitively challenging tasks (for work and recreation), social interactions, creative activities (including artistic, musical and written works), and so forth. However, any change (e.g. enhancement) in cognitive stimulation will be relative to the cognitive activities and engagement at baseline. Attempts to formulate and implement cognitive training (or 'brain training' as it is often referred to in the public domain) needs to be scaled to the cognitive abilities and inclinations of individuals, to ensure sufficient novelty and complexity, and continued cognitive challenges and engagement over time. In laboratory rodents (and other animal species), there is the experimental challenge of ensuring that the 'cognitively stimulated' group engages in the same amount of physical activity, and therefore the careful design of cognitive stimuli and control groups is crucial in order to distinguish the effects of cognitive stimulation from those of physical activity, and their combination (which is generally defined as environmental enrichment). This has been best illustrated in rodent models of Huntington's disease (e.g. Hockly et al., 2002; van Dellen et al., 2000; Wood et al., 2011) and Alzheimer's disease (e.g. Anderson et al., 2017; Arendash et al., 2004; Billings et al., 2007; Brendborg & Febbraio, 2026; Jankowsky et al., 2005; Lazarov et al., 2005; Martinez-Coria et al., 2015; Rai et al., 2020; Shepherd et al., 2018; Shepherd et al., 2021).

## Epimimetics

Epimimetics are a subclass of enviromimetics that target epigenetic effects of environmental stimulation, in order to mimic or enhance their beneficial effects (Hannan, 2020). Due to constraints of space and their prior detailed description (Hannan, 2020), the present article will only provide a brief discussion of epimimetics.

The most important consideration as that many of the long-term beneficial effects of environmental stimulation (including via physical activity and cognitive stimulation) are likely to involve epigenetic mechanisms. Epigenetics allows complex molecular changes to be encoded in specific cells over very long periods of time (Oh & Petronis, 2021). This means that epigenetics can facilitate enormous amounts of 'information storage' in cells that can be used in response to beneficial environmental stimuli (Hannan, 2020). Therefore the subclasses of exercise mimetics and cognitomimetics are almost certain to overlap with epimimetics, as epigenetic modifications are prime targets of environmental stimuli.

Furthermore, epigenetics plays a major role in a wide range of human disorders, so that both 'epigenopathy' and 'epigenetic reliance' are important concepts when

considering new strategies for disease prevention and treatment. Epimimetics may therefore provide a novel approach which utilizes the known epigenetic impacts of therapeutic interventions such as exercise and cognitive stimulation (Bittel and Chen, 2024; García-Giménez et al., 2024; Gomez-Pinilla & Thapak, 2024; Hannan, 2020; Zhang et al., 2024). A key research question involves the kinds of environmentally induced epigenetic modifications that would be likely targets for epimimetics. This may include a wide range of DNA modifications (including DNA methylation) and histone modifications, as well as non-coding RNAs (both small and long non-coding RNAs). Epigenetics is regulated by molecular 'writers', 'readers' and 'erasers'. All of these kinds of molecules are potential targets for epimimetics; however, one of the major challenges will be to attain therapeutic specificity whilst minimizing potential side effects (Hannan, 2020). Whilst these challenges are non-trivial, the potential therapeutic benefits justify substantial investment and further research.

## Conclusions

The vast majority of neurological and psychiatric disorders remain either untreated, or poorly treated, thus leaving a very substantial unmet need in this area of science and medicine. This is partly because of our poor understanding of their pathogenesis and associated biological mechanisms. However, another reason may be that approaches to the development of new treatments have been mainly conservative and focused on traditional approaches to drug development. There is therefore an urgent need, based on this enormous collective burden of disease, to explore new approaches, driven by the latest scientific evidence informing the aetiology of these disorders and associated pathogenic mechanisms. Enviromimetics, and their key sub-classes including exercise mimetics, provide one such approach that can complement the more traditional drug-discovery programmes that are currently being pursued by academia and industry (see Abstract Figure). Considering the extraordinary, and growing, global burden of brain disorders, new approaches to prevention and treatment are an urgent priority for researchers, clinicians, governments and industry.

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

## Additional information

### Competing interests

None declared.

### Author contributions

Sole author.

### Funding

Federal Government | DHAC | National Health and Medical Research Council (NHMRC): Anthony Hannan, 2 032 021; Federal Government | DHAC | National Health and Medical Research Council (NHMRC): Anthony Hannan, 2 023 333.

### Acknowledgements

The author thanks the past and present members of the Hannan Laboratory for experimental data and discussions that have informed the ideas presented in this article. AJH has been supported by grants from the NHMRC, including an NHMRC Principal Research Fellowship, Ideas Grant and co-funded grants with EU-JPND and ERA NET NEURON, as well as the DHB Foundation (Equity Trustees), Flicker of Hope Foundation and the Margaret Friend Trust.

Open access publishing facilitated by The University of Melbourne, as part of the Wiley - The University of Melbourne agreement via the Council of Australian University Librarians.

### Keywords

drug discovery, enviromimetics, exercise mimetics, environmental enrichment, neurological diseases, physical activity, psychiatric disorders, therapeutics

## Supporting information

Additional supporting information can be found online in the Supporting Information section at the end of the HTML view of the article. Supporting information files available:

**Peer Review History**

