## [Peer Review History · The Journal of Physiology]

Enviromimetics: From exercise mimetics to cognitomimetics in the quest for enhanced brain health and cognition

Anthony Hannan

DOI: 10.1113/JP287484

Corresponding author(s): Anthony Hannan (anthony.hannan@florey.edu.au)

The following individual(s) involved in review of this submission have agreed to reveal their identity: Emily N Copeland (Referee #1); Jessica L Braun (Referee #2)

Review Timeline:

Submission Date:	02-Mar-2025
Editorial Decision:	19-Mar-2025
Revision Received:	19-Aug-2025
Accepted:	03-Sep-2025

Senior Editor: Laura Bennet

Reviewing Editor: Rebecca MacPherson

Transaction Report:

Dear Dr Hannan,

Re: JP-TR-2025-287484 "**Enviromimetics: From exercise mimetics to cognitomimetics in the quest for enhanced brain health and cognition**" by Anthony Hannan

Thank you for submitting your manuscript to The Journal of Physiology. It has been assessed by a Reviewing Editor and by 2 expert referees and we are pleased to tell you that it is potentially acceptable for publication following satisfactory major revision.

Please address all the points raised and incorporate all requested revisions or explain in your Response to Referees why a change has not been made. We hope you will find the comments helpful and that you will be able to return your revised manuscript within 2 months. If you require longer than this, please contact journal staff: jp@physoc.org. Please note that this letter does not constitute a guarantee for acceptance of your revised manuscript.

ABSTRACT FIGURES: Authors are expected to use The Journal's premium BioRender account to create/redraw their Abstract Figures. Information on how to access this account is here:

<https://physoc.onlinelibrary.wiley.com/journal/14697793/biorender-access>.

REVISION CHECKLIST:

IMPORTANT POINTS TO NOTE WHEN REVISING YOUR MANUSCRIPT:

LANGUAGE EDITING AND SUPPORT FOR PUBLICATION: If you would like help with English language editing, or other article preparation support, Wiley Editing Services offers expert help, including English Language Editing, as well as translation, manuscript formatting, and figure formatting at www.wileyauthors.com/eoo/preparation. You can also find resources for Preparing Your Article for general guidance about writing and preparing your manuscript at www.wileyauthors.com/eoo/prepresources.

We look forward to receiving your revised submission.

Yours sincerely,

Laura Bennet
Senior Editor
The Journal of Physiology

REQUIRED ITEMS

- Please include an Abstract Figure file, as well as the Figure Legend text within the main article file. The Abstract Figure is a piece of artwork designed to give readers an immediate understanding of the Review Article and should summarise the main conclusions. If possible, the image should be easily 'readable' from left to right or top to bottom. It should show the physiological relevance of the Review so readers can assess the importance and content of the article. Abstract Figures should not merely recapitulate other figures in the Review. Please try to keep the diagram as simple as possible and without superfluous information that may distract from the main conclusion of the Review. Abstract Figures must be provided by authors no later than the revised manuscript stage and should be uploaded as a separate file during online submission labelled as File Type 'Abstract Figure'. Please ensure that you include the figure legend in the main article file. All Abstract Figures will be sent to a professional illustrator for redrawing and you may be asked to approve the redrawn figure before your paper is accepted.

- Author profile(s) must be uploaded via the submission form. Authors should submit a short biography (no more than 100 words for one author or 150 words in total for two authors) and a portrait photograph of the two leading authors on the paper. These should be uploaded and clearly labelled together in a Word document with the revised version of the manuscript. Any standard image format for the photograph is acceptable, but the resolution should be at least 300 DPI and preferably more. A group photograph of all authors is also acceptable, providing the biography for the whole group does not exceed 150 words.

EDITOR COMMENTS

Reviewing Editor:

The review provides a comprehensive summary of enviromimetics, including exercise mimetics and epimimetics, while introducing cognitomimetics as a novel therapeutic class. However, the manuscript lacks detail and specificity, particularly in linking enviromimetics to brain health and cognition. To enhance clarity, several improvements are suggested: cognitive stimulation should be further defined with examples like puzzles, learning new skills, or creative activities; a summary at the end of the myokines section would help explain how these factors benefit brain health and cognition. Additionally, introducing BDNF as a central neurotrophic factor affected by exercise-induced myokines, providing specifics on how molecules are activated by exercise, and including examples of microRNA, sncRNA, and lncRNA are recommended. The reviewers also suggests reorganizing sections for better relevance. Finally, clarifying the statement about reducing side effects of exercise mimetics, particularly whether these are brain-specific, would provide a more comprehensive understanding of the challenges in developing these treatments. Overall, these adjustments would improve the clarity and depth by adding specific details and examples.

Key Points are not required for a Topical Review.

Please also see 'Required Items' above.

REFEREE COMMENTS

Referee #1:

In this review, the author provides a summary of enviromimetics, including exercise mimetics and the proposal of a new subtype, cognitomimetics. The manuscript was interesting and provided a new outlook on the future of therapeutics relating to enviromimetics. Though thought provoking, the manuscript lacked details and specificity, throughout and the relation of enviromimetics to brain health and cognition, as suggested in the title, seemed to be lost. My suggestions of improvement are as follows:

1. In the myokines as molecular targets of exercise mimetics subsection, many of these myokines (line 194-217) relate to BDNF. To create a better flow of text, the author may wish to introduce BDNF as a neurotrophic factor that can be acted upon by numerous myokines following exercise to promote brain health.

I. Additionally, the author may wish to include how each of these myokines could be targeted and how they are activated through exercise.

II. A reference may be required for line 193.

III. Further explanations should be included as to the types of cognitive performance that is being enhanced (ex. line 202) and specific results from the cited research should be included for clearer understanding.

2. Similar to the suggestions listed above, the other molecules that are candidate targets for exercise mimetics subsection should include further specifics to each molecule suggested. For example, how they are activated by exercise, which pathways it acts upon, how this connects to cognition, etc.

I. A number of therapeutics were mentioned on lines 245 and 246, but these treatments should be explained to their full extent. How did these treatments act on these pathways? Were the results beneficial or not? How do these improve cognition, if they do?

II. The author should include examples of the "specific" microRNA, sncRNA, and lncRNA that they mention throughout lines 226-291. Much of this information could be merged and revised to save space.

3. The author may wish to move components of the cellular mediators of the therapeutic effects of exercise subsection to previous sections where this information might be more relevant as opposed to having its own subsection. Furthermore, the brain-gut axis and EV transport information is interesting but does not necessarily fit with the theme of this manuscript. Very little of this ties into the myokines and exerkinines that were previously mentioned and therefore adds little to the manuscript itself. I suggest removing.

4. Within the Cognitomimetics section, the proposal itself is very interesting. However, the author should consider including more tangible suggestions to go along with this proposal. For example, what types of cognitive stimulation? Learning to play a new instrument? Building a puzzle? And how might these act on similar or different molecular targets that were previously mentioned?

5. In the Epimimetics section, the author mentions space as being a constraint to not define epimimetics. However, without this definition, this section seems irrelevant. To save space, I will recommend the author determine if this adds to the article, or if this space could be used elsewhere to build on suggestions or include more detailed background.

Referee #2:

The review by A. Hannan provided a summary of current knowledge surrounding environmimetics, specifically exercise mimetics and epimimetics, with current and hypothesized mechanisms. A new class called cognitomimetics was also proposed providing an interesting new avenue of research and making significant contributions to the field. The review is well-structured and there are a few points that would increase clarity and improve it further.

1. It would be beneficial to further define cognitive stimulation (lines 94-96) and provide some examples, especially in the context of what has been shown to enhance brain health.

2. The summary of the work by Horowitz et al 2020 could be reworded to increase clarity. For example, it is unclear to the reader that the circulating blood factors from exercised aged mice were given to sedentary aged mice, improving brain health (lines 158-162).

3. The author could consider adding a summary for the reader at the end of the myokines section (starting line 166) describing how these myokines released following exercise can be beneficial for brain health and cognition in health and disease states.

4. It may be worth expanding briefly on the targeting of the AMPK-SIRT1-PGC1-PPAR pathway for brain health (paragraph lines 241-248) as was done in the following paragraphs

5. The author may wish to describe further what is meant by "reduce their side effects" (line 345). Is this referring to side effects of treating with exercise mimetics? Are these brain-specific side effects?

END OF COMMENTS

I greatly appreciate the constructive comments from the Editor and the two Referees. I have extensively revised the manuscript to address each and every one of the comments, and attach the fully revised manuscript. Below I detail the responses to the comments.

EDITOR COMMENTS

Reviewing Editor:

The review provides a comprehensive summary of enviromimetics, including exercise mimetics and epimimetics, while introducing cognitomimetics as a novel therapeutic class. However, the manuscript lacks detail and specificity, particularly in linking enviromimetics to brain health and cognition. To enhance clarity, several improvements are suggested: cognitive stimulation should be further defined with examples like puzzles, learning new skills, or creative activities; a summary at the end of the myokines section would help explain how these factors benefit brain health and cognition. Additionally, introducing BDNF as a central neurotrophic factor affected by exercise-induced myokines, providing specifics on how molecules are activated by exercise, and including examples of microRNA, sncRNA, and lncRNA are recommended. The reviewers also suggests reorganizing sections for better relevance. Finally, clarifying the statement about reducing side effects of exercise mimetics, particularly whether these are brain-specific, would provide a more comprehensive understanding of the challenges in developing these treatments. Overall, these adjustments would improve the clarity and depth by adding specific details and examples.

RESPONSE: I have addressed all of these comments. Further details are provided below, in my individual responses to each of the Referees.

REFEREE COMMENTS

Referee #1:

In this review, the author provides a summary of enviromimetics, including exercise mimetics and the proposal of a new subtype, cognitomimetics. The manuscript was interesting and provided a new outlook on the future of therapeutics relating to enviromimetics. Though thought provoking, the manuscript lacked details and specificity, throughout and the relation of enviromimetics to brain health and cognition, as suggested in the title, seemed to be lost. My suggestions of improvement are as follows:

1. In the myokines as molecular targets of exercise mimetics subsection, many of these myokines (line 194-217) relate to BDNF. To create a better flow of text, the author may wish to introduce BDNF as a neurotrophic factor that can be acted upon by numerous myokines following exercise to promote brain health.

RESPONSE: The text of the revised manuscript has been revised to address these constructive comments.

I. Additionally, the author may wish to include how each of these myokines could be targeted and how they are activated through exercise.

RESPONSE: Some additional text has been added to address this excellent point.

II. A reference may be required for line 193.

RESPONSE: This has been added to the revised manuscript.

III. Further explanations should be included as to the types of cognitive performance that is being enhanced (ex. line 202) and specific results from the cited research should be included for clearer understanding.

RESPONSE: This further explanation has been provided in the text of the revised manuscript.

2. Similar to the suggestions listed above, the other molecules that are candidate targets for exercise mimetics subsection should include further specifics to each molecule suggested. For example, how they are activated by exercise, which pathways it acts upon, how this connects to cognition, etc.

I. A number of therapeutics were mentioned on lines 245 and 246, but these treatments should be explained to their full extent. How did these treatments act on these pathways? Were the results beneficial or not? How do these improve cognition, if they do?

RESPONSE: These are helpful suggestions, and text has been added to address these points.

II. The author should include examples of the "specific" microRNA, sncRNA, and lncRNA that they mention throughout lines 226-291. Much of this information could be merged and revised to save space.

RESPONSE: I have added specific microRNA and lncRNA examples, and associated citations and references.

3. The author may wish to move components of the cellular mediators of the therapeutic effects of exercise subsection to previous sections where this information might be more relevant as opposed to having its own subsection. Furthermore, the brain-gut axis and EV transport information is interesting but does not necessarily fit with the theme of this manuscript. Very little of this ties into the myokines and exerkinines that were previously mentioned and therefore adds little to the manuscript itself. I suggest removing.

RESPONSE: I greatly appreciate this constructive suggestion but feel that the cellular mediators section should stay separate. I also feel that the gut-brain axis and EV transport information is within the theme of the manuscript. But I do acknowledge that the flow of the manuscript could be improved, so have gone through the entire manuscript to improve structure and flow, to address these constructive comments.

4. Within the Cognitomimetics section, the proposal itself is very interesting. However, the author should consider including more tangible suggestions to go along with this proposal. For example, what types of cognitive stimulation? Learning to play a new instrument? Building a puzzle? And how might these act on similar or different molecular targets that were previously mentioned?

RESPONSE: I have addressed this excellent suggestion, to provide some examples and specifics.

5. In the Epimimetics section, the author mentions space as being a constraint to not define epimimetics. However, without this definition, this section seems irrelevant. To save space, I will recommend the author determine if this adds to the article, or if this space could be used elsewhere to build on suggestions or include more detailed background.

RESPONSE: This is a highly constructive suggestion and I have extended the Epimimetics section to provide more details.

Referee #2:

The review by A. Hannan provided a summary of current knowledge surrounding environmimetics, specifically exercise mimetics and epimimetics, with current and hypothesized mechanisms. A new class called cognitomimetics was also proposed providing an interesting new avenue of research and making significant contributions to the field. The review is well-structured and there are a few points that would increase clarity and improve it further.

1. It would be beneficial to further define cognitive stimulation (lines 94-96) and provide some examples, especially in the context of what has been shown to enhance brain health.

RESPONSE: This constructive comment has been addressed.

2. The summary of the work by Horowitz et al 2020 could be reworded to increase clarity. For example, it is unclear to the reader that the circulating blood factors from exercised aged mice were given to sedentary aged mice, improving brain health (lines 158-162).

RESPONSE: Text has been added to fully address this excellent suggestion.

3. The author could consider adding a summary for the reader at the end of the myokines section (starting line 166) describing how these myokines released following exercise can be beneficial for brain health and cognition in health and disease states.

RESPONSE: This constructive comment has been incorporated.

4. It may be worth expanding briefly on the targeting of the AMPK-SIRT1-PGC1-PPAR pathway for brain health (paragraph lines 241-248) as was done in the following paragraphs

RESPONSE: Text and citations/references have been added to address this informed suggestion.

5. The author may wish to describe further what is meant by "reduce their side effects" (line 345). Is this referring to side effects of treating with exercise mimetics? Are these brain-specific side effects?

RESPONSE: A sentence has been added to address this excellent suggestion. These constructive comments are greatly appreciated.

Dear Professor Hannan,

Re: JP-TR-2025-287484R1 "**Enviromimetics: From exercise mimetics to cognitomimetics in the quest for enhanced brain health and cognition**" by Anthony Hannan

We are pleased to tell you that your paper has been accepted for publication in The Journal of Physiology.

Authors should note that it is too late at this point to offer corrections prior to proofing. Major corrections at proof stage, such as changes to figures, will be referred to the Editors for approval before they can be incorporated. Only minor changes, such as to style and consistency, should be made at proof stage. Changes that need to be made after proof stage will usually require a formal correction notice.

Yours sincerely,

Laura Bennet
Senior Editor
The Journal of Physiology

P.S. - You can help your research get the attention it deserves! Check out Wiley's free Promotion Guide for best-practice recommendations for promoting your work at www.wileyauthors.com/eeo/guide. You can learn more about Wiley Editing Services which offers professional video, design, and writing services to create shareable video abstracts, infographics, conference posters, lay summaries, and research news stories for your research at www.wileyauthors.com/eeo/promotion.

IMPORTANT NOTICE ABOUT OPEN ACCESS: To assist authors whose funding agencies mandate public access to published research findings sooner than 12 months after publication, The Journal of Physiology allows authors to pay an Open Access (OA) fee to have their papers made freely available immediately on publication.

You can check if your funder or institution has a Wiley Open Access Account here: <https://authorservices.wiley.com/author-resources/Journal-Authors/licensing-and-open-access/open-access/author-compliance-tool.html>.

EDITOR COMMENTS

Reviewing Editor:

The authors have addressed all previous concerns.

REFeree COMMENTS

Referee #1:

I have read the revised manuscript and response letter and am satisfied with the revisions and responses.

Referee #2:

The author has addressed all comments.